# Lessons from Post-Immunotherapy Tumor Tissues in Clinical Trials: How Can We Fuel the Tumor Microenvironment in Gliomas?

**DOI:** 10.3390/vaccines12080862

**Published:** 2024-08-01

**Authors:** Lan Hoc Phung, Takahide Nejo, Hideho Okada

**Affiliations:** 1Department of Neurological Surgery, University of California, San Francisco, CA 94143, USA; lan.phung@ucsf.edu (L.H.P.); takahide.nejo@ucsf.edu (T.N.); 2Helen Diller Family Comprehensive Cancer Center, University of California, San Francisco, CA 94158, USA; 3Parker Institute for Cancer Immunotherapy, San Francisco, CA 94129, USA

**Keywords:** glioblastoma, glioma, neoadjuvant treatment, immunotherapy, clinical trials, trial designs, tumor microenvironment

## Abstract

Despite recent advancements in cancer immunotherapy, many patients with gliomas and glioblastomas have yet to experience substantial therapeutic benefits. Modulating the tumor microenvironment (TME) of gliomas, which is typically “cold”, is crucial for improving treatment outcomes. Clinical tumor specimens obtained post-immunotherapy provide invaluable insights. However, access to such post-immunotherapy samples remains limited, even in clinical trials, as tumor tissues are often collected only at tumor relapse. Recent studies of neoadjuvant immunotherapy provided important insights by incorporating surgical resections of post-treatment tumors. Moreover, pre-surgical immunotherapies are increasingly integrated into clinical trial designs to evaluate treatment efficacy. These investigations reveal critical information, particularly regarding the delivery success of therapeutic agents, the expansion and persistence of immune products, and the cellular and molecular changes induced in the TME. In this review, we assess the findings on post-treatment tumor specimens obtained from recent immunotherapy clinical trials on gliomas, highlight the importance of these samples for understanding therapeutic impacts, and discuss proactive investigation approaches for future clinical trials.

## 1. Introduction

Gliomas, particularly glioblastomas (GBMs), are among the most aggressive and treatment-resistant primary central nervous system (CNS) tumors. Despite advances in surgical and medical treatments, patients continue to face dismal prognoses. Over the past decade, cancer immunotherapy, represented by immune checkpoint blockade (ICB) and chimeric antigen receptor (CAR) T-cell therapy, has revolutionized cancer treatment for hematological malignancies and melanomas [1,2,3]. However, many patients with GBMs have yet to experience appreciable therapeutic benefits [4,5,6,7]. The brain’s immune-privileged status, characterized by the blood–brain barrier and a highly immunosuppressive environment, significantly contributes to the treatment resistance observed in these diseases [8,9].

Although many new immunotherapeutic options showed promising results in preclinical and early-phase clinical trials, many failed to demonstrate meaningful clinical benefits in later-phase trials [10]. Numerous factors are thought to contribute to these failures, including the insufficient delivery of therapeutic agents and suboptimal modulation of the TME being significantly challenging. To better understand the causes of treatment failures and the precise resistance mechanisms, it is crucial to evaluate the actual effects of treatments on the TME [10,11].

In clinical trials, tumor tissue collection is usually performed before the actual study treatment (pre-treatment). To evaluate therapeutic efficacy, less invasive methods are preferentially employed over surgeries, such as imaging tests, blood tests (including the characterization of peripheral blood mononuclear cells [PBMCs]), and sometimes cerebrospinal fluid (CSF) tests. These methods, however, only provide indirect insights into the consequences of the therapies. Ultimately, the true status of the TME can be thoroughly inspected only through post-treatment tumor tissue specimens.

Evaluating these post-treatment tumor tissues presents several challenges. From an ethical standpoint, performing tumor biopsies solely for investigative purposes is difficult to justify, given the potential risks to patients without direct clinical benefits [12]. As a result, post-treatment tumor tissues are not always available and are typically obtained only at tumor relapse or progression, at autopsy, or when there is a clinical indication. A sampling of recurrent tumors after immunotherapy failure is not ideal due to the inconsistency in the timing of sampling among the patients and the lack of opportunities to learn about immune responses linked with clinical benefits. Despite these challenges, post-treatment tumor tissues provide critical insights into how treatments impact the TME, including information on the success of therapeutic agent delivery, immune cell infiltration, and the status of these immune cells.

In this review, we summarize recent clinical trials investigating immunotherapies in gliomas, focusing on the key lessons learned from the analysis of post-treatment tumor tissues. We then discuss the importance of information obtained from post-treatment tumor tissues and the approaches for future clinical trial designs.

## 2. Methodology

We conducted a systematic literature search using PubMed to identify relevant studies on 8 May 2024. The search phrase used was: (“neoadjuvant” OR “vaccine” OR “CAR T” OR “T cell” OR “vaccination” OR “immunotherapy” OR “checkpoint inhibitor”) AND (“glioma” OR “glioblastoma” OR “IDH mutation” OR “IDH1”) AND (“2019” [Date—Publication]: “3000” [Date—Publication]). From the initial results list, we applied the following filter: Article Type as Clinical Trial. This refined search yielded 84 results, from which we manually curated articles based on the following criteria: studies exclusively involving brain tumors and those involving post-treatment tumor tissue analyses. The final selection consisted of 23 studies, including 14 that analyzed post-treatment (without a scheduled surgical resection) tumor tissues at relapse (Table 1) and 9 that analyzed tumor tissues in neoadjuvant (pre-surgical) treatment settings (Table 2).

### 2.1. Analysis Data of Post-Treatment Tumor Tissue at Disease Progression

As previously mentioned, most clinical trials only collected and analyzed post-treatment samples upon tumor relapse [13,14,15,18,19,22,23,25,26]. Typically, the number of samples analyzed was limited [13,15,20,22,24,25,26], and the timing of tissue collection post-treatment varied among subjects, even within the same studies [13,15,19,23,25,26]. Furthermore, a sampling at recurrence does not provide opportunities to learn about immune responses linked with clinical response. Therefore, the findings are often hard to generalize and remain speculative [13,14,18,19,20,21,22,23,24,25,26] Nevertheless, the analysis of post-treatment tumor tissues has yielded valuable insights, such as evidence of treatment agent penetration [13,15,16,17,18,19,21,22,23,24,25,26], changes in TME [13,14,15,16,17,18,19,20,21,22,23,24,25,26], or lack of expected changes [14,15,16,18,20,21,24,25,26]. The studies below best exemplify these points.

For instance, in an open-label, multi-institutional phase I trial, Chiocca et al. (2022) evaluated the safety and immunobiological effects of combining veledimex (VDX)-regulated interleukin-12 (IL-12) gene therapy with an immune checkpoint blockade (ICB) using nivolumab in recurrent GBM (rec-GBM) [17]. Twenty-one patients received nivolumab and VDX seven days and three hours before tumor resection, respectively. Liquid chromatography-mass spectrometry (LC-MS) analysis demonstrated effective VDX penetration into brain tumor tissue with a dose–response relationship. Four patients underwent additional tumor resections for suspected recurrence among the study subjects. Immunohistochemistry (IHC) and multiplexed immunofluorescence revealed a significant decrease in PD-1+ and PD-L1+ cells post-treatment, indicating reduced immune checkpoint expression. However, this was unexpectedly accompanied by a reduction in activated tumor-infiltrating lymphocytes (TILs) between pre- and post-treatment tissues. The authors speculated that this might be due to the timing of nivolumab administration with IL-12 gene therapy or a potential deleterious effect of neoadjuvant anti-PD-1 on immune activation.

The unexpected reduction in activated TILs post-treatment in this study suggests potential negative interactions or suboptimal timing between the therapies. The combination of VDX-regulated IL-12 gene therapy and nivolumab might lead to conflicting immune signals, where IL-12 aims to boost immune responses, but the timing or dose of nivolumab might prematurely inhibit these responses. This finding emphasizes the necessity of carefully timing combination therapies to avoid counterproductive effects. The consistent collection of tumor tissues at pre-determined time points could provide critical insights into the optimal sequencing and dosing of such combinations, enabling a more precise modulation of the TME to enhance therapeutic efficacy.

Similarly, Ling et al. (2023) investigated the effects of CAN-3110, an oncolytic herpes simplex virus (oHSV), on 41 patients with rec-GBM [23]. In this open-label, single-institution, phase I trial, post-treatment tumor tissues were primarily collected at disease progression and in post-mortem re-resections of 23 patients. PCR confirmed the presence of CAN-3110-specific viral DNA in tumor tissues, indicating successful viral replication and spreading. Histology and immunohistochemistry analyses revealed an increased infiltration of CD4+ and CD8+ tumor-infiltrating lymphocytes, suggesting an enhanced immune response within the TME. T-cell-receptor beta (TCRβ) DNA sequencing showed increased TCRβ diversity post-treatment, associated with prolonged survival, indicating a broad and effective anti-tumor immune response. RNA sequencing (RNA-seq) identified a highly inflammatory and immunologically activated TME in HSV1 serologically positive patients.

These findings suggest that CAN-3110 effectively infects GBM cells and elicits a robust immune response, transforming the typically “cold” glioma microenvironment into a more immunologically active state. However, the reliance on post-mortem and progression samples may limit our understanding of the therapy’s full impact during earlier stages. For this reason, the consistent collection of tumor tissues at pre-determined time points will be beneficial to gain a comprehensive understanding of the temporal dynamics of immune activation and the full therapeutic potential of CAN-3110.

In another study, Bagley et al. (2024) conducted a phase I trial to test repeated peripheral infusions of anti-EGFRvIII CAR T cells combined with pembrolizumab in patients with newly diagnosed EGFRvIII+ GBM [25]. While the treatment was found to be safe and well-tolerated, the expansion and persistence of infused T cells were minimal, and no objective clinical efficacy was observed. All seven patients underwent repeat surgery for suspected disease progression at various times. Consistent with the group’s prior study, EGFRvIII levels in the pre- and post-treatment tumor tissues revealed a decrease in the target antigen in six of seven patients [35], although there is a known tendency for the EGFRvIII mutation to be lost over time, even in the absence of EGFR-targeting therapy [36]. Using 4-1BB/CD3z (BBZ) quantitative PCR (qPCR), the authors detected CAR T cells in the blood of five out of seven patients at the time of repeated tumor resections. In contrast, CAR T cells were detected in the tumor tissue of only one patient, who had the shortest interval between CAR T-cell infusion and tumor tissue collection. This patient also exhibited the highest levels of CAR T cells in peripheral circulation at this time point, suggesting that CAR T-cell infiltration into tumors may occur during peak peripheral expansion. However, it remains unclear if other patients experienced similar infiltration. Interestingly, no CAR T cells were detected in scRNA-seq analysis, even in this patient, highlighting the lower sensitivity of this modality. The varying durations from the last CAR T-cell infusion to tumor resection may have limited our understanding of the kinetics of the infused cells and the resultant modulation of the TME.

This study provided important insights into the challenges of CAR T-cell therapy in GBM, particularly regarding the limited persistence and penetration of CAR T cells in the TME. These observations underscore the need for strategies to improve CAR T-cell infiltration and persistence within the TME. Thus, the consistent collection of tumor tissues at pre-determined time points could help track the dynamics of CAR T-cell presence and activity, offering valuable information on how to enhance their effectiveness and overcome resistance mechanisms.

Overall, these studies highlight the complexities and challenges of understanding and improving the effectiveness of immunotherapies for GBM. The variability in sample collection times and reliance on post-mortem or progression samples emphasize the need for more structured and consistent approaches to tissue collection, which would allow for a more detailed and accurate understanding of how these therapies modulate the TME over time and across different treatment stages.

### 2.2. Analysis Data of Post-Treatment Tumor Tissues at a Predefined Timing

In contrast to the studies discussed in the previous section, incorporating predefined timing for tissue collection offers more consistent and insightful data. For instance, the feasibility and efficacy of the neoadjuvant (pre-surgical) administration of checkpoint inhibitors to treat patients with GBM were investigated in two separate clinical trials, offering consistent opportunities for tissue collection.

Cloughesy et al. (2019) conducted a randomized, multi-institution clinical trial on patients with recurrent, surgically resectable GBM to evaluate the immune responses and survival benefits of neoadjuvant and adjuvant pembrolizumab (*n* = 16 patients) compared to adjuvant treatment alone (*n* = 16 patients) [27]. RNA-seq analysis revealed that neoadjuvant PD-1 blockade upregulated T-cell- and interferon-γ-related gene expression while downregulating cell-cycle-related gene expression within the tumor. Immunofluorescence indicated that neoadjuvant anti-PD-1 treatment was associated with the focal induction of PD-L1 expression and high CD8+ T-cell infiltration. TCR sequencing demonstrated that neoadjuvant anti-PD-1 uniquely initiated a coordinated local and systemic T-cell response, which was not observed with adjuvant treatment alone. These findings highlight the positive effects of neoadjuvant pembrolizumab.

This study illustrated the significant immunomodulatory effects of administering pembrolizumab in the neoadjuvant setting for patients with recurrent GBM. Collecting tumor tissues at pre-determined time points during neoadjuvant therapy provided critical insights into the optimal timing and sequencing of treatments, potentially enhancing therapeutic efficacy and improving patient outcomes.

Similarly, Schalper et al. (2019) conducted a single-arm phase II clinical trial to evaluate the feasibility, safety, and immunobiological effects of neoadjuvant nivolumab in patients with resectable GBM [28]. Gene expression profiling of tumor samples showed an upregulation of numerous immune-related transcripts in nivolumab-treated tumors compared with historical control samples. Flow cytometry revealed that most CD8+ effector cells expressed activation markers, such as CD69 and HLA-DR. Intriguingly, differential staining using multiple monoclonal antibodies (competitive and non-competitive with nivolumab binding to PD-1) confirmed at least the partial receptor occupancy of PD-1 on these cells. TCR sequencing demonstrated increased clonal T-cell diversity following neoadjuvant nivolumab treatment compared to paired pre-treatment and control series samples. Multiplexed immunofluorescence indicated that nivolumab treatment resulted in minimal changes or mild increases in immune cell markers, whereas standard treatment led to a global reduction in both adaptive and innate immune cells, indicating the preventive effect of nivolumab.

This study underscored the immunomodulatory benefits of neoadjuvant nivolumab in resectable GBM. The upregulation of immune-related transcripts and increased activation of CD8+ effector cells suggest an enhanced anti-tumor immune response. The observed partial receptor occupancy of PD-1 on T cells and increased clonal T-cell diversity indicate the effective engagement and activation of the immune system. However, the minimal changes in immune cell markers in the tumor highlight the need for further investigations into the timing and sequencing of neoadjuvant treatments. Regardless, collecting tumor tissues at pre-determined time points provided deeper insights into the optimal therapeutic windows and improved our understanding of the immune dynamics, ultimately enhancing treatment efficacy.

These findings from post-treatment tumor tissue assessments highlight the significant immunomodulatory effects of neoadjuvant pembrolizumab and nivolumab on the TME and underscore the importance of assessing post-treatment tumor tissues. Such assessments are invaluable for fully understanding the impact and potential benefits of the investigated treatment options.

Further supporting the importance of pre-determined time points for tumor tissue collection, Todo et al.’s (2022) investigator-initiated, phase 2 single-arm trial assessed the efficacy of G47Δ, triple-mutated oncolytic herpes simplex virus type 1, in patients with residual or recurrent supratentorial GBMs [32] G47Δ was administered intratumorally, with patients receiving up to six doses repeatedly. The study’s design allowed multiple injections and multi-time-point tumor biopsies from different coordinates each time. Biopsies confirmed that none of the recurrent cases was pseudoprogression. The study observed an increase in CD4+ and CD8+ T-cell infiltration following repeated G47Δ injections, along with persistently low numbers of Foxp3+ cells. These findings suggest a correlation between Foxp3+ cells and G47Δ efficacy. This multi-injection approach provided valuable longitudinal data, highlighting the dynamic modulation of the TME that could impact therapeutic efficacy.

This study demonstrated the benefits of repeated administrations and multi-coordinate biopsies in capturing the dynamic modulation of the TME. Foxp3+ cells are typically regulatory T cells (Tregs) [37] which play a role in maintaining immune tolerance and suppressing excessive immune responses [38]. In the context of cancer, Tregs can inhibit the activity of effector T cells, such as CD8+ cytotoxic T cells and CD4+ helper T cells, thereby allowing the tumor to evade the immune system [39] A reduction in Foxp3+ cells can relieve this suppression, allowing effector T cells to function more effectively. The increased T-cell infiltration and reduced Foxp3+ cells following repeated G47Δ injections suggest a favorable shift toward an anti-tumor immune environment. Collecting tumor tissues at pre-determined time points provided a clearer understanding of how these changes evolve over time, informing strategies for sustained therapeutic efficacy and better patient outcomes.

Moreover, two recent studies reported the neoadjuvant administration of vaccines to patients with low-grade gliomas (LGGs). Ogino et al. (2022) conducted a randomized pilot study to evaluate the immunological effects of a neoadjuvant vaccine composed of GBM6-AD lysate and poly-ICLC in patients with WHO grade 2 LGGs [31]. In this study, patients received the vaccine before the surgical resection of their tumors. TCRβ sequencing revealed that certain TCRβ clonotypes enriched in post-vaccinated peripheral blood were also found in the corresponding tumor tissue, indicating the successful migration of vaccine-reactive T cells into the TME. Additionally, mass cytometry showed a significantly higher proportion of CD103+CD8+ T cells with an effector memory phenotype in tumors from the neoadjuvant vaccine group, with an increased expression of the chemokine receptor CXCR3. These findings suggest that the neoadjuvant vaccination can induce a robust peripheral and intratumoral immune response.

The findings underscore the potential of neoadjuvant vaccination strategies to enhance immune cell infiltration and activity within the glioma TME. The detection of vaccine-reactive T-cell clonotypes in both peripheral blood and tumor tissues highlights the importance of these cells’ migration and persistence in the TME. Moreover, the increased presence of effector memory CD8+ T cells in the tumor, particularly those expressing CXCR3, suggests a more effective immune environment post-vaccination. This study supports the argument that analyzing post-treatment tumor tissues can provide crucial insights into the efficacy of immunotherapeutic strategies.

In another study, Saijo et al. (2023) conducted a prospective, randomized pilot study evaluating the efficacy of the multi-peptide IMA950 vaccine combined with varlilumab, an agonistic anti-CD27 antibody, in patients with CNS WHO grade 2 LGGs [33]. In this study, 10 patients received 4 cycles of neoadjuvant vaccines with or without varlilumab, followed by tumor resection. Flow cytometry with HLA tetramer staining revealed significant increases in anti-IMA950 CD8+ T-cell responses in peripheral blood induced by the neoadjuvant vaccines. However, no IMA950-reactive CD8+ T cells were detected in the resected tumor tissues, indicating a failure of immune cells to infiltrate the tumor. Additionally, mass cytometry showed that varlilumab promoted effector memory T-cell differentiation in peripheral blood but not within the TME, suggesting that the TME of LGG may not be permissive to the peripherally induced immune responses. The discrepancies observed between peripheral blood and tumor tissues underscore the importance of assessing post-treatment tumor tissues to fully understand the impact of therapies.

This study revealed a critical challenge in glioma immunotherapy: the discrepancy between peripheral immune responses and those within the TME. Despite robust peripheral activation, the lack of corresponding intratumoral responses indicates barriers to T-cell infiltration or survival within the tumor. Collecting tumor tissues at pre-determined time points helped identify these discrepancies and provided insights into stages at which these barriers arise, thus offering a clearer understanding of how to modify the TME to support effective immune cell infiltration and function.

Together, these studies demonstrate the critical role of timing and methodical tissue collection in advancing our understanding of immunotherapeutic strategies for GBM. By systematically analyzing post-treatment tumor tissues at predefined intervals, researchers gained valuable insights into the dynamic changes in the TME induced by various treatments, particularly about tumoral gene expression changes and immune cell infiltration and activation [27,28,31,32,33,34,40]. Furthermore, integrating meticulous tissue collection into clinical trial designs enhances our ability to optimize timing and intervals for the treatment and evaluation of immune responses [27,28,29,30,31,33,34,40], ultimately strengthening our ability to maximize therapeutic efficacy and improve patient outcomes.

## 3. Discussion

As reviewed in this article, post-treatment tumor tissue samples provide invaluable insights into the effectiveness of treatment agents, including their penetration, distribution, and persistence, as well as potential changes induced within the TME and unexpected treatment resistance mechanisms. Discrepancies often arise between findings from tumor tissues and those from less invasively acquired specimens, such as blood and CSF [31,33]. For instance, favorable changes observed in PBMCs post-treatment are not always mirrored by the TME, indicating challenges related to penetration into the lesions, which can potentially explain treatment failures. Post-treatment tumor tissues are the most informative data source for accurately characterizing treatment consequences within the TME and improving disease management.

Moreover, collecting these tissues at uniform, predefined time points, rather than randomly at relapse, would enable sophisticated analyses with statistical power, ultimately benefiting future patients. By integrating such approaches into future clinical trial designs, researchers can better understand and optimize the conditions for effective immune cell infiltration and activity within the glioma TME. Advanced tools and techniques, such as multiplexed immunofluorescence, RNA-seq, flow cytometry, and TCR profiling, provide detailed insights into the cellular and molecular compositions of the TME (Figure 1). Utilizing these tools, as demonstrated by the studies mentioned earlier, can help elucidate the treatment responses of immunotherapies, allowing for the development of more effective treatment strategies.

Despite the importance of this research design, the benefit of tissue collection for study participants is still debated [12] Patients with brain tumors typically undergo post-treatment surgeries only when clinically indicated, such as for tumor mass reduction or diagnostic purposes (e.g., confirming recurrence or suspected pseudoprogression) [41] Without such indications, these invasive procedures generally do not directly benefit patients and pose certain risks. Consequently, justifying these procedures, known as research biopsies [12] is challenging, especially in brain tumor settings.

From this perspective, neoadjuvant treatments [27,28,31,33] and multi-session treatment agent deliveries [29,32] offer promising solutions. These approaches do not compromise patient benefits while providing in-depth and relatively well-controlled information on post-treatment tumor tissues at uniform time points. Indeed, some recent or ongoing clinical trials have also incorporated this methodology [42]. Additionally, predefined tumor biopsies solely to assess treatment response could be considered. If specific clinical decisions, such as the continuation or discontinuation of treatment, can be made based on tissue analysis, the procedure would be ethically justifiable and directly beneficial to the patient. To achieve this, reliable biomarkers must be established. Since peripheral surrogate markers (e.g., changes in PBMCs) do not always accurately reflect the TME’s response, efforts should be made to collect and analyze post-treatment tumor tissues without compromising patient benefits.

Through the analysis of post-treatment tumor tissues, researchers can gain invaluable insights into the dynamic interactions within the TME. Of note, a framework for standardized tissue sampling and processing has recently been discussed, which is important for maximizing information retrieval from the collected specimens [43]. Future clinical trial designs should incorporate strategies to maximize the availability of post-treatment tissues without compromising patient safety. Ultimately, integrating these strategies will refine immunotherapeutic approaches, paving the way for more effective and personalized treatments to treat future patients with brain tumors.

## 4. Conclusions

The analysis of post-treatment tumor tissues offers crucial insights into the mechanisms and efficacy of glioma immunotherapies. However, the variability in sample collection times, especially when relying on disease progression or post-mortem, is a key limitation for fully understanding the dynamic changes within the TME post-treatment. Therefore, predefined timing for tissue collection in clinical trials provides a reliable solution, allowing for more precise assessments of immunotherapies in the TME. This approach ultimately expands our understanding of the intricacies within the TME and improves patient outcomes. Future clinical trials should consider the incorporation of predefined tissue collection without compromising patient safety.

## Figures and Tables

**Figure 1 vaccines-12-00862-f001:**
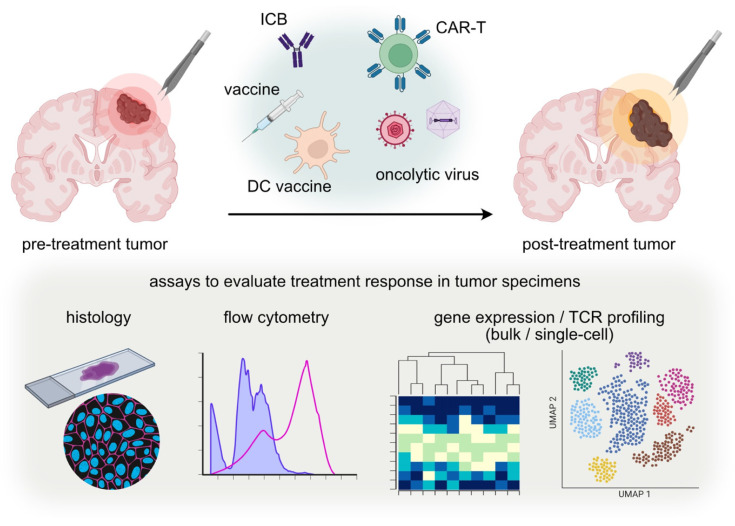
Evaluating treatment response through the analyses of post-treatment tumor tissue specimens.

**Table 1 vaccines-12-00862-t001:** Reported clinical trials with analysis data of post-treatment tissue at disease progression.

Author, Year	Trial Registration	Disease	Treatment Investigated	*n* (Patients)	Key Assays and Findings
Chiocca EA, 2019 [13]	NCT02026271 (ATI001-102)	Rec-GBM or anaplastic astro (grade 3)	Regulatable interleukin-12 gene therapy (tumor cavity injection)	5	4 out of 5 patients were found to be pseudoprogression. IHC/IF: increased tumor-infiltrating lymphocytes producing IFNγ and PD-1. Intratumoral IFNγ concentration increased after gene therapy.
Hilf N, 2019 [14]	NCT02149225 (GAPVAC-101)	ND-GBM	Personalized peptide vaccines (unmutated, or unmutated and mutated)	1	IHC: high infiltration by T cells with a favorable CD8+ T/FOXP3+ Treg cell ratio. CD4+ T-cell response: the tumor contained CD4+ T cells directed against the APVAC1 pan-DR peptide PTP-010.
Keskin, 2019 [15]	NCT02287428 (NeoVax trial)	MGMT-unmethylated, ND-GBM	Personalized neoantigen vaccines	5	Multiplexed IF: a significant increase in infiltrating CD8+ T cells at relapse in 2 patients, whereas no increase was observed in the other 3 patients who received dexamethasone. TCR repertoire seq: a subset of putatively reactive TCR α and β-chain sequences were directly detectable in the post-treatment sample RNA, suggesting the successful trafficking to the site of disease. scRNA-seq: nearly all CD8+ T cells appeared to be differentiated cells (CCR7–) and expressed markers of cytotoxicity (PRF1, GZMA, and GZMK).
Migliorini D, 2019 [16]	NCT01920191	ND-GBM and grade 3 astrocytoma	IMA950 multi-peptide vaccine and poly-ICLC	7	IHC: no major changes in antigen expression were observed in recurrent samples. No correlation was observed between tumor antigen expression and antigen-specific CD8+ T-cell responses.
Chiocca EA, 2021 [17]	NCT03636477	Rec-GBM	Ad-RTS-hIL-12: Veledimex (VDX)-regulatable IL-12 gene therapy with neoadjuvant nivolumab (anti-PD-1 Ab)	4	LC-MS: a dose–response relationship with effective brain tumor tissue VDX penetration was observed. Histology/multiplexed-IF: a significant decrease in the number of PD-1+ cells and PD-L1+ cells was observed. The addition of ICBs reduced PD-1/PD-L1 expression. Activated TILs also decreased between pre-and post-treatment tissues.
Duerinck J, 2021 [18]	NCT03233152	Rec-GBM	Preoperative nivolumab and peritumoral administration of nivo or nivolumab + ipilimumab	3	Histology: no evidence of tumor recurrence, but immune cell infiltration was observed in 2 out of 3 tumors.
Friedman GK, 2021 [19]	NCT02457845	pediatric HGG	G207: Oncolytic HSV-1	4	IHC: HSV-1 staining was completely negative in any post-treatment tissues, which indicated that G207 was no longer replicating. A brisk infiltration of CD8+ T cells and increases in CD20+ B-cells and CD138+ plasma cells were revealed.
Platten M, 2021 [20]	NCT02454634 (NOA16)	ND-, WHO grades 3 and 4 IDH1(R132H)+ astrocytoma	IDH1-vac: IDH1(R132H)-specific peptide vaccine	1	ELISPOT: IDH1(R132H)-reactive T cells were identified from lesion-infiltrating leukocytes (LILs). scRNA-seq/scTCR-seq: among the three clusters of CD4+ T cells, two non-regulatory T-cell clusters were dominated by one TCR (“TCR14”). TCR14 was enriched 50.6-fold in the PsPD lesion compared to the patient’s PBMC.
Brown CE, 2022 [21]	NCT01082926	Rec-GBM	GRm13Z40-2 cells: healthy-donor-derived IL13Rα2-targeted CAR T cell	2	IHC: IL13Rα2 expression was maintained, and CD8+ T-cell infiltration increased. FISH: only limited numbers of GRm13Z40-2 cells persisted since treatment.
Gállego Pérez-Larraya J, 2022 [22]	NCT03178032	Pediatric, ND-DIPG	DNX-2401: an oncolytic adenovirus	1	Multiplex IF: at relapse, increases in CD8+ and CD4+ T cells and a decrease in myeloid cells were observed. In contrast, reductions in CD8+ and CD4+ T cells and an increase in CD163+ M2 macrophages were observed. sn-RNA-seq: after treatment, tumor-infiltrating macrophages showed upregulations of viral process and immune response pathways.
Ling AL, 2023 [23]	NCT03152318	Rec-GBM	CAN-3110: an oncolytic herpes virus (oHSV)	29	PCR: the presence of CAN-3110-specific viral DNA was confirmed. Histology/IHC: increases in CD8+ and CD4+ TILs. TCRβ-DNA-seq: increased tumor TCRβ diversity was associated with prolonged post-treatment survival. RNA-seq: a highly inflammatory and immunologically activated TME in HSV1 serologically positive patients.
Liu Z, 2023 [24]	NCT03170141	GD2+, Rec-or progressive GBM	GD2-specific 4S-CAR T cells	1	IHC/IF: GD2 antigen loss and T-cell infiltration were observed.
Bagley SJ, 2024 [25]	NCT03726515	EGFRvIII+, ND-GBM	Anti-EGFRvIII-CAR T with pembrolizumab (anti-PD-1 Ab).	7	qPCR: only in 1 out of 7 tumors and CAR T cells were detected in the brain via BBZ qPCR. scRNA-seq: no CAR T cells were found, including in the qPCR-positive case. Increases in exhaustion markers and IFN-stimulated signatures were observed after treatment.
Choi BD, 2024 [26]	NCT05660369 (INCIPIENT study)	EGFRvIII+, ND- or Rec-GBM	CARv3-TEAM-E T cells: EGFRvIII-CAR also secreting T-cell-engaging antibody molecules [TEAM] against wt-EGFR	1	NGS and IHC: negative for EGFRvIII, while a gain in the EGFR copy number was maintained.

ND-, newly diagnosed-; rec-, recurrent-; HGG, high-grade glioma; DIPG, diffuse intrinsic pontine glioma.

**Table 2 vaccines-12-00862-t002:** Reported clinical trials with analysis data of post-treatment tissue at predefined timing.

Author, Year	Trial Registration	Disease	Treatment Investigated	*n* (Patients)	Time from Treatment	Key Assays and Findings
Cloughesy TF, 2019 [27]	N/A	Rec-GBM	Neoadjuvant anti-PD-1 (pembrolizumab) (vs. adjuvant only)	14	14 days +/− 5	RNA-seq: upregulation of T-cell- and interferon-γ-related gene expressions, but downregulation of cell-cycle-related gene expression within the tumor. IF: neoadjuvant anti-PD-1 treatment is associated with focal induction of PD-L1 expression with a high CD8 infiltrate. TCR-seq: neoadjuvant anti-PD-1 uniquely initiated a coordinated local and systemic T-cell response.
Schalper KA, 2019 [28]	NCT02550249	ND- and rec-GBM	Neoadjuvant nivolumab (anti-PD-1 Ab)	30	14 days +/− 3	Nanostring: Nivo-treated samples showed an upregulation of numerous immune-related transcripts. FCM: most CD8+ eff cells expressed CD69 and HLA-DR, indicating activation and/or tissue residence. IHC (in 3 cases): confirmed at least partial receptor occupancy at the time of surgery, as revealed by differential staining treatments using mAbs targeting PD-1 (in 3 cases). TCR-seq: increased clonal T-cell diversity following neoadjuvant Nivo treatment. Multiplexed IF: Nivo treatment was associated with a minimal change or a mild increase in immune cell markers, whereas the standard treatment (control) was associated with a global reduction in both adaptive and innate immune cell indicators.
Weathers SP, 2020 [29]	NCT02661282	ND- and rec-GBM	Autologous CMV pp65-specific T cells	1	8 days	ELISA: CD8+ T cells isolated from GBM-TME were more refractory to stimulation and unreactive to CMV-peptide stimulation. IHC: CMV-specific CD8+ T cells were PD-1 positive, mostly in the tumor vasculature and not spreading, indicating they were dysfunctional.
Kasenda B, 2022 [30]	NCT03603379 (GBM-LIPO trial)	EGFR-amplified, Rec-GBM	Anti-EGFR ILs-dox: anti-EGFR immunoliposomes loaded with doxorubicin	3	24 h	PK: doxorubicin was detectable in the tumor tissues 24 h after treatment, whereas it was undetectable in CSF. IHC/IMC: CD68+ macrophage population was relatively more frequent in two patients after treatment, while a clear reduction, along with a lower proliferation of glioma cells, was observed in the other patient.
Ogino H, 2022 [31]	NCT02549833	ND- or rec-WHO grade 2 gliomas	GBM6-AD: allogeneic cell lysate-based vaccine	13	2 days after 4 cycles of vaccines	TCRβ-seq: some TCRβ clonotypes enriched in post-vaccinated peripheral blood were also identified in the corresponding tumor tissue, suggesting the successful migration of the vaccine-reactive T cells into the TME. Mass cytometry: the proportion of CD103+CD8+ T cells with an effector memory phenotype was significantly higher in tumors in the neoadjuvant vaccine group, with a higher positivity for CXCR3.
Todo T, 2022 [29]	UMIN000002661	Rec- or progressive GBM	G47∆: a triple-mutated oncolytic HSV type 1	13	0	IHC: decreased number of tumor cells, infiltration of CD4+ and CD8+ T cells, and HSV-1 positive staining were observed.
Todo T, 2022 [32]	UMIN000015995	Rec- or residual GBM	G47∆: a triple-mutated oncolytic HSV type 1	19 (3)	0	IHC: it was confirmed that all recurrent cases were not pseudoprogression. Increased numbers of CD4+ and CD8+ T-cell infiltration and persistent low numbers of Foxp3+ cells were observed. At tumor progression, increased numbers of Foxp3+ cells were found in the two cases at 4 months.
Saijo A, 2023 [33]	NCT02924038	WHO grade 2 LGG	IMA950 multi-peptide vaccine ± varlilumab (agonistic anti-CD27 Ab)	10	2 days after 4 cycles of vaccines	Mass cytometry: adding varlilumab induced detectable changes in PBMCs but not in the TME.
Galanis E, 2024 [34]	NCT00390299	Rec-GBM	CEA-MV: CEA-expressing oncolytic measles virus derivative	11 (5)	5 days	Nanostring: the gene expression scores of interferon-stimulated genes were inversely correlated with virus replication.

## Data Availability

Not applicable.

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
