# Peer review of "Lessons from Post-Immunotherapy Tumor Tissues in Clinical Trials: How Can We Fuel the Tumor Microenvironment in Gliomas?"

_vaccines, 2024, doi:10.3390/vaccines12080862_

Round 1

Reviewer 1 Report

Comments and Suggestions for Authors

In this review, the authors aim to show that post-treatment tissue analysis can provide valuable data to better understand the efficacy outcomes of therapeutic options studied in recurrent glioblastoma, and thus guide future therapies. To this end, they carried out a systematic search of clinical trials on recurrent glioblastoma in which post-treatment tumor specimens were analyzed.  Using various examples, the authors list the information extracted from these analyses in terms of distribution, persistence, penetration of therapy, local or systemic immunomodulatory effect or remodeling of the microenvironment thanks to new imaging and/or omic analysis technologies with single-cell resolution. They also highlight discrepancies between what is observed from post-treatment tumor samples and samples such as blood or cerebrospinal fluid. Nevertheless, as the authors point out, the number of studies, the samples analyzed in each study, the diversity of sample recovery and processing protocols, and the timing of sampling all constitute important limitations, and the results are difficult to generalize and remain speculative for the most part.

The authors could have put more emphasis on the importance of establishing guidelines or consensus statements on how best to collect and manage post-treatment tumor specimens for maximal information retrieval.

This review raises more questions than it answers - how to manage ethical issues, patient benefits and insights into the clinical efficacy of a therapy- but the subject is topical and will be of interest to readers. It is well written and easy to read.

Author Response

We thank the reviewer for the thoughtful review and encouraging words about our manuscript. In response to the reviewer’s suggestions, we have added the following sentence to the Discussion section.

Page 11

Of note, a framework for standardized tissue sampling and processing has recently been discussed, which is important for maximizing information retrieval from the collected specimens [43].

ref 43. Karschnia, P., Smits, M., Reifenberger, G., Le Rhun, E., Ellingson, B. M., Galldiks, N., Kim, M. M., Huse, J. T., Schnell, O., Harter, P. N.; et al. A framework for standardised tissue sampling and processing during resection of diffuse intracranial glioma: joint recommendations from four RANO groups. Lancet. Oncol. 2023, 24, e438–e450

Reviewer 2 Report

Comments and Suggestions for Authors

This review paper is well-written and organized. The authors highlight the importance of analyzing post-treatment tumor specimens to understand treatment efficacy and resistance mechanisms which is crucial for advancing cancer therapies. 

One minor comment is that it would be better to include subtitles for each study example or use double lines for separation to improve readability. 

Author Response

Thank you for the careful review of our work and your encouraging comments. Following your suggestion, we have reformatted the manuscript by inserting double lines in the section on study examples. We believe this has improved readability.

Reviewer 3 Report

Comments and Suggestions for Authors

This article does not move the field of immunotherapy forward but is a bleak view of brain research and how it is behind the rest of the immunotherapy field that has accepted window of opportunity studies in neoadjuvant therapy.  One issue is the definition of "neoadjuvant" therapy. Neoadjuvant means before surgery in oncology terms therefore if there is no surgery but only therapy it is not "neoadjuvant". Would re-write to define pre-surgery therapy (second half) and therapy without surgery (first half). The results would be the same but it would be more clinically relevant.   Otherwise hopefully will drive more studies that do collect tissue even for brain research.

Comments on the Quality of English Language

The language in this article does need some revising. There is considerable repetition (for example of However starting multiple sentences) and some of the sentences are run on. This is particularly in the introduction and discussion, once discussing the studies the writing is much clearer

Author Response

Thank you for the careful review of our work and your encouraging comments. The term “neoadjuvant” in our writing agrees with the reviewer’s comment. We agree that neoadjuvant means before surgery; therefore, if there is no surgery, and it is the same intent we use in the paper. In response to the reviewer’s suggestions, we have clarified our language usage in the Methodology section. We have also revised our sentences in the Introduction and Discussion sections as you recommended.

Page 2

The final selection consisted of 23 studies, including 14 that analyzed post-treatment (without scheduled surgical resection) tumor tissues at relapse (Table 1) and 9 that an-alyzed tumor tissues in neoadjuvant (pre-surgical) treatment settings (Table 2).